# The Nuclear Envelope in Lipid Metabolism and Pathogenesis of NAFLD

**DOI:** 10.3390/biology9100338

**Published:** 2020-10-15

**Authors:** Cecilia Östlund, Antonio Hernandez-Ono, Ji-Yeon Shin

**Affiliations:** 1Department of Medicine, Vagelos College of Physicians and Surgeons, Columbia University, New York, NY 10032, USA; co69@cumc.columbia.edu (C.Ö.); ah2003@cumc.columbia.edu (A.H.-O.); 2Department of Pathology and Cell Biology, Vagelos College of Physicians and Surgeons, Columbia University, New York, NY 10032, USA

**Keywords:** nuclear envelope, nuclear membrane-associated protein, NAFLD, NASH, lipid droplet, TorsinA, LAP1, VLDL, apoB

## Abstract

**Simple Summary:**

The liver is a major organ regulating lipid metabolism and a proper liver function is essential to health. Nonalcoholic fatty liver disease (NAFLD) is a condition with abnormal fat accumulation in the liver without heavy alcohol use. NAFLD is becoming one of the most common liver diseases with the increase in obesity in many parts of the world. There is no approved cure for the disease and a better understanding of disease mechanism is needed for effective prevention and treatment. The nuclear envelope, a membranous structure that surrounds the cell nucleus, is connected to the endoplasmic reticulum where the vast majority of cellular lipids are synthesized. Growing evidence indicates that components in the nuclear envelope are involved in cellular lipid metabolism. We review published studies with various cell and animal models, indicating the essential roles of nuclear envelope proteins in lipid metabolism. We also discuss how defects in these proteins affect cellular lipid metabolism and possibly contribute to the pathogenesis of NAFLD.

**Abstract:**

Nonalcoholic fatty liver disease (NAFLD) is a burgeoning public health problem worldwide. Despite its tremendous significance for public health, we lack a comprehensive understanding of the pathogenic mechanisms of NAFLD and its more advanced stage, nonalcoholic steatohepatitis (NASH). Identification of novel pathways or cellular mechanisms that regulate liver lipid metabolism has profound implications for the understanding of the pathology of NAFLD and NASH. The nuclear envelope is topologically connected to the ER, where protein synthesis and lipid synthesis occurs. Emerging evidence points toward that the nuclear lamins and nuclear membrane-associated proteins are involved in lipid metabolism and homeostasis. We review published reports that link these nuclear envelope proteins to lipid metabolism. In particular, we focus on the recent work demonstrating the essential roles for the nuclear envelope-localized torsinA/lamina-associated polypeptide (LAP1) complex in hepatic steatosis, lipid secretion, and NASH development. We also discuss plausible pathogenic mechanisms by which the loss of either protein in hepatocytes leads to hepatic dyslipidemia and NASH development.

## 1. Introduction

Nonalcoholic fatty liver disease (NAFLD) is a condition with accumulation of excess fat in the liver without significant alcohol consumption. NAFLD is a wide spectrum of diseases, ranging from mild steatosis, which is lipid droplet (LD) deposition in hepatocytes, to nonalcoholic steatohepatitis (NASH) that is characterized by steatosis, inflammation, and ballooning degeneration of hepatocytes [1]. Simple hepatic steatosis is benign and reversible; however, NASH is a potentially serious condition that can further develop to liver cirrhosis and hepatocellular carcinoma (HCC). NAFLD/NASH is frequently associated with metabolic comorbidities such as type 2 diabetes, cardiovascular disease, and chronic kidney disease, which make the management of patients with the disease challenging [2]. NAFLD is one of the most common causes of liver disease in both adults and adolescents in the United States and is found in up to 25–30% of the population. Approximately 10–20% of people with NAFLD develop its advanced stage, NASH [3]. During the last two decades, the prevalence of NAFLD has more than doubled with the high prevalence of obesity in developed countries, and it is rapidly being recognized as the leading cause of chronic liver disease; however, there is no approved treatment for NASH [4].

The lack of an effective treatment for NASH is due in part to the incomplete understanding of the pathogenesis of NAFLD and NASH, which is hampered by the complexity of the disorders. Growing evidence suggests that NASH is a multifaceted process that requires the convergence of diverse genetic, metabolic, and environmental factors [5,6,7,8,9]. Numerous pharmaceutical and biotechnology companies are devoting significant efforts to better understand NASH and develop treatments. Data from phase II and phase III clinical trials show that none of the current drugs in clinical development for NASH provide robust responses in a majority of patients [10]. Therefore, multiple and novel lines of investigation need to be undertaken to discover major susceptibility factors and potential druggable targets for NASH treatment. 

Despite the complexity of the pathogenesis, one constant feature of NAFLD/NASH is that it always develops in the context of hepatic steatosis, the abnormal LD accumulation in hepatocytes. When various intracellular and extracellular stress events are combined with the steatosis, hepatocytes undergo stress/damage responses and ultimately, cellular injury and death, key features of NASH, which are associated with inflammation and metabolic disturbance. Therefore, comprehensive understanding of hepatocyte intrinsic factors that link lipid metabolism to various cellular stress responses is pivotal to understand the pathophysiology of NAFLD/NASH. 

In this review, we will describe recent findings and knowledge regarding the emerging novel connection between the nuclear envelope and lipid metabolism based on studies using different model systems. The nuclear envelope is physically connected to the endoplasmic reticulum (ER), the main organelle for cellular lipid synthesis. The nuclear envelope proteins are involved in diverse and essential cellular functions and mutations in genes encoding these proteins cause multiple human diseases involving various tissues [11,12]. Growing evidence indicates that the nuclear envelope is a metabolically active site and that nuclear lamins and other nuclear membrane-associated proteins play important roles in lipid metabolism, which can contribute to the development of NAFLD and NASH. In particular, we will highlight a recent report with compelling evidence showing that a protein complex localized in the nuclear envelope is a novel mediator of steatosis and NASH. We will then discuss possible cellular pathways for how defects in nuclear membrane-associated proteins lead to abnormal hepatic lipid metabolism and speculate on the contribution of these proteins to the development of NAFLD/NASH.

## 2. Nuclear Envelope and Lipid Metabolism 

The nuclear envelope separates the nucleoplasm from the cytoplasm in nucleated cells. It consists of a double membrane; the outer nuclear membrane is contiguous with the ER and shares many protein and features with the main ER. The inner nuclear membrane has a distinct set of proteins and is lined with a filamentous network, the nuclear lamina, which consists of A-type and B-type lamin proteins [13] (Figure 1). The inner and outer nuclear membranes are connected at the nuclear pore complexes, through which soluble proteins carrying nuclear localization signals can be actively transported, and proteins smaller than 40 kDa can diffuse [14]. While the nuclear envelope originally was thought to be a passive partition between the nucleus and the rest of the cell, it has, during recent years, been shown to play an active role in various processes, and mutations in nuclear envelope proteins are associated with many different diseases. Diseases caused by mutations in the lamin filament proteins and proteins associated with them, sometimes referred to as Laminopathies, includes myopathies, cardiomyopathies, neuropathies, multisystem disorders with features of premature aging, such as progeria, and lipodystrophies [13,15]. Recently, several studies have also implicated the nuclear envelope in playing active roles in lipid metabolism. These roles can be played directly through lipid metabolism enzymes situated at the nuclear envelope, by nuclear envelope proteins indirectly affecting transcription of lipidomic genes, or by a combination of these processes [16,17,18,19].

The ER is the main site of lipid synthesis in the cell, with most lipid synthesis enzymes being transmembrane proteins [17]. As the outer nuclear membrane and perinuclear space are contiguous with the ER membrane and lumen, it is not surprising that lipid synthesis occurs in the nuclear envelope as well. As it is challenging to purify nuclear envelopes without ER contaminants, the role of nuclear membranes in this process has not been well understood. During recent years, however, the identification of nuclear envelope proteins involved in lipid metabolism, as well as of mutations in nuclear envelope proteins causing lipid abnormalities, has pointed towards an active role of the nuclear envelope in lipid metabolism. It is not known if the lipid composition of the inner nuclear membrane is distinct from that of the ER. As the lipid membranes are contiguous, lipids are believed to freely diffuse between them, but could be enriched or regulated in the inner nuclear membrane by the concentration of proteins involved in lipid metabolism in the nuclear envelope [20].

### 2.1. Nuclear Lipid Droplets

Lipid droplets consist of neutral lipids, like triacylglycerols and cholesterol esters, coated by a phospholipid monolayer and proteins. They act as lipid reservoirs for energy production and membrane biogenesis, and are now seen as dynamic organelles that can interact with other organelles in a regulated manner [18,21]. Lipid droplets are found in the cytoplasm of many cells, but recent studies have also identified lipid droplets in the nucleus in several cell types, like hepatocytes and hepatocyte-derived cell lines [19,22]. Normally relatively sparse even in these cells, an increase in nuclear lipid droplets (nLDs) has been seen in different pathological conditions as well as in mice fed high-fat diets, hepatocytes from mice lacking the inner nuclear membrane protein LAP1, and cells grown with an addition of oleic acid [21,23,24]. Although nLDs share many features with cytoplasmic lipid droplets (cLDs), their lipid and protein contents are unique [22]. The differential impacts of nLDs and cLDs on the NAFLD pathogenesis and progression are unknown.

The role of the inner nuclear membrane in lipid metabolism and nLD formation is poorly understood. In hepatocyte-derived cell lines, nLDs have been found to be associated with promyelocytic leukemia (PML) bodies and the type I nucleoplasmic reticulum (NR), invaginations of the inner nuclear membrane into the nuclear interior [25]. A recent study in hepatocytes showed that nLDs are derived from apolipoprotein B (apoB)-free VLDL precursors in the ER lumen. During VLDL formation, lipidated apoB and apoB-free VLDL particles fuse to generate mature VLDL, which is then secreted. During ER stress, apoB levels decrease, leaving apoB-free VLDL particles trapped in the ER/nuclear envelope lumen. These then form large lipid droplets in the type I NR, which enter the nucleoplasm through defects in the inner nuclear membrane [19]. The formation and function of type I NR is not well understood, but hepatocytes under ER stress exhibit an increase in type I NR. While type II NR consists of invaginations of both the outer and inner nuclear membrane, type I NR only contains inner nuclear membrane. This would require a separation of the two membranes, which normally are connected by nuclear pore complexes and the linker of nucleoskeleton and cytoskeleton (LINC) complex. In agreement with this model, knockdown of SUN1, a LINC complex protein, induces an increase in type I NR [25]. In budding yeast (*Saccharomyces cerevisiae*), nLDs are generated by the inner nuclear membrane by a yeast-specific mechanism. This mechanism does not involve type 1 NR; instead, the nLDs are generated by triglyceride synthesis in the inner nuclear membrane [19,26].

### 2.2. Nuclear Lamins and Membrane Proteins Implicated in Lipid Metabolism

We review recent studies that provide the link between the nuclear envelope proteins and lipid metabolism in various experimental systems. Some nuclear membrane-residing or associated proteins contain enzyme activities that can directly catalyze lipid synthesis or modifying reactions. Some of the mutations in genes encoding these proteins lead to diseases attributed to abnormal lipid metabolism.

#### 2.2.1. Lamin B receptor (LBR)

The Lamin B receptor (LBR) is a transmembrane protein of the inner nuclear membrane. The amino-terminal region of the protein, which faces the nucleoplasm, associates with the nuclear lamina. The carboxy-terminal region contains eight transmembrane domains and has sequence homology to sterol C14 reductases [27,28,29]. LBR mutations have been implicated in two congenital disorders: Pelger–Huët anomaly, where a single mutation in one LBR allele causes hypolobulated granulocyte nuclei [30] and Greenberg skeletal dysplasia, which is a recessive, perinatally lethal, bone development disease [31]. Studies of mutations causing Greenberg skeletal dysplasia showed them to affect residues evolutionarily conserved between sterol reductases. Mutant protein failed to rescue the cholesterol auxotrophy of LBR-deficient cell lines, showing that LBR is essential for cholesterol synthesis, despite humans having a second sterol reductase called TM7SF2, located in the ER [32,33].

#### 2.2.2. Lipins and CTDNEP1/NEP1R1

Lipins are a family of phosphatidic acid phosphatases, which are localized to both the nucleus and the cytoplasm. They play an important role in lipid metabolism by dephosphorylating phosphatidic acid (PA) to diacylglycerol (DAG). While PA is a precursor to phosphatidylinositol, DAG is used for synthesis of phosphatidylcholine, phosphatidylethanolamine, and triacylglycerol. Lipins in mammalian cells also have roles as transcriptional coactivators of PPAR transcription factors, which promote fatty acid oxidation. In their active, dephosphorylated state, lipins are associated with membranes, while phosphorylation induces membrane dissociation [34,35]. Dephosphorylation of lipins is carried out by C-terminal domain nuclear envelope phosphatase 1 (CTPNEP1), which is an evolutionary conserved transmembrane protein mainly localized to the nuclear envelope, in a physical complex with nuclear envelope phosphatase 1-regulatory subunit 1 (NEP1-R1) [36,37]. A recent study by Jacquemyn et al. has shown that *Drosophila* torsin is an upstream regulator of this pathway [38]. It negatively regulates the NEP1R1–CTDNEP1 complex by preventing accumulation of CTDNEP in the nuclear envelope, which in turn prevents dephosphorylation and activation of lipin, inhibiting conversion of PA to DAG. Phosphorylation of lipins is carried out by mTORC1. Co-expression of CTDNEP1 and NEP1-R1, as well as inhibition of mTORC1, increase the nuclear fraction of lipin 1, which induces changes to the shape of the nucleus. The molecular details of this deformation of the nucleus are not well understood and could involve both signaling and structural mechanisms [35,39,40]. Changes in lipid content in the inner nuclear membrane due to abnormal lipin activity could also induce invaginations such as type 1 NR [35].

#### 2.2.3. Choline-Phosphate Cytidylyltransferase A (PCYT1A)

PCYT1A, which catalyzes the rate-limiting step for phosphatidylcholine production via the Kennedy pathway, has, similarly to lipin, an inactive, phosphorylated, soluble state and an active, dephosphorylated, membrane-associated state. While inactive PCYT1A has a nucleoplasmic localization, it is translocated to the nuclear envelope, NR, and/or cytosol when stimulated with oleate [41]. Both lipin and PCYT1A have nuclear localization signals, which enable them to shuttle between the nucleus and cytosol via the nuclear pore complexes. 

#### 2.2.4. A-Type Lamins

Although most laminopathy mutations identified so far affect muscle tissue, early studies of mutations in A-type lamins also identified mutations causing Dunnigan-type Familial Partial Lipodystrophy (FPLD2) [42,43]. This disease is characterized by loss of subcutaneous fat from the extremities, trunk, and gluteal region as well as metabolic issues such as insulin resistance, central obesity, hyperinsulinemia, glucose intolerance and diabetes, dyslipidemia, hypertension, and hepatic steatosis, which may progress to steatohepatitis [44,45,46]. Common in other lipodystrophies, hepatic steatosis occurs in patients with FPLD2 mainly secondary to the defects in adipose tissues. To further understand the role of lamins in the liver, Kwan et al. created mice with a hepatocyte-specific deletion of A-type lamins. These mice have abnormally shaped hepatocyte nuclei and male mice develop spontaneous hepatosteatosis, which progress to steatohepatitis and fibrosis when the mice are fed a high-fat diet. Expression profiling showed a male-specific interference with growth hormone Stat5 signaling and an activation of genes related to steatosis, inflammation, and fibrosis, as well as an increased Stat1 expression and signaling [47]. The authors concluded that A-type lamins act cell-autonomously in hepatocyte homeostasis and suggested that proteins are potential genetic modifiers for predisposition to steatohepatitis and fibrosis.

#### 2.2.5. LAP2

Sibling studies have revealed variants of genes encoding nuclear envelope proteins, which confer a significantly increased risk of NAFLD. Over one-third of variants identified in a study by Brady et al. were in the *TMPO* gene, which encodes lamina-associated polypeptide 2 (LAP2) [48]. One of these variants resulted in a truncated form of LAP2 (aa 1–99), which when expressed in Huh7 cells increased lipid accumulation compared to the expression of full-length LAP2. It was also mislocalized, had unique biochemical properties, and altered the endogenous lamin distribution in Huh7 cells [48]. This study is the first to associate LAP2 variants with NAFLD and is a further indication of the importance of the nuclear lamina in this disease.

#### 2.2.6. Lamin B1

Autosomal dominant leukodystrophy (ADLD), a fatal adult onset demyelination disorder, is caused by gene duplication of *LMNB1* or by genomic deletions upstream of the *LMNB1* gene [49,50]. Both genetic alterations induce the overexpression of lamin B1 in oligodendrocytes, specialized myelin producing cells in the central nervous system [51]. In a mouse model, increased lamin B1 levels in oligodendrocytes caused the destruction of myelin in the spinal cord, which was mediated by reduced gene expression of lipid synthesis genes such as 3-hydroxyl-3-methylglutaryl-CoA synthase 1 (Hmgcs1) and sterol regulatory binding protein (SREBP) 1 and 2, which are important for myelin formation and maintenance. The authors also found that lamin B1 overexpression causes a reduction in myelin-enriched lipid species such as cholesterol and phospholipid, which can lead to demyelination in mice [52]. These results establish the crucial role for the nuclear lamin B in lipid biosynthesis in myelin-producing oligodendrocytes.

## 3. The TorsinA/LAP1 Complex in Lipid Metabolism and NASH Development

A recent study by Shin and colleagues has demonstrated that the torsinA/lamina-associated polypeptide 1 (LAP1) complex at the nuclear envelope regulates hepatic lipid secretion and steatosis [24]. Along with other recent reports mentioned above, these results underscore the crucial roles for the nuclear envelope and nuclear membrane-associated proteins in hepatic lipid homeostasis and the nuclear envelope as a novel cellular site linked to NAFLD/NASH development. The connection between the torsinA/LAP1 complex and lipid metabolism was identified unexpectedly, as most previous research on the torsinA/LAP1 complex had been centered on neuronal tissue and striated muscles due to the human diseases caused by mutations in the genes encoding torsinA or LAP1. We will briefly review previous studies on the protein complex and recent findings on the connection of the complex to lipid metabolism and its implication in NAFLD/NASH development.

### 3.1. The Discovery of LAP1 and TorsinA Interaction and Its Implications in Human Diseases

LAP1 was originally identified in rat liver nuclear envelope protein extracts in 1988 [53]. Further biochemical extraction showed that it is an integral protein localized to the inner nuclear membrane and associated with nuclear lamins [54]. The rodent LAP1 encoding gene, *Tor1aip1*, is expressed in three isoforms (LAP1A, LAP1B, and LAP1C), with apparent molecular masses of 75, 68, and 55 kDa [53]. These isoforms are expressed in most tissues in a differential pattern dependent on the developmental stages [55]. In humans, at least two isoforms are expressed from the *TOR1AIP1* gene [56]. LAP1 is a type II integral protein with nucleoplasmic, transmembrane, coiled-coil, and luminal domains [57] (Figure 2A). After its identification as a nuclear lamina-associated protein, LAP1 has been reported to interact with multiple proteins. The nucleoplasmic domain of LAP1 interacts with nuclear lamins and emerin, another integral protein of the inner nuclear membrane [58]. Other studies identified different binding partners of LAP1, including protein phosphatase 1 and telomeric repeat-binding factor 2 [59,60,61]. 

One of the most studied LAP1 interacting proteins is torsinA [62]. Torsins are ATPase-associated proteins with diverse cellular activities (AAA+). There are four torsins (torsinA, torsinB, torsin2A, and torsin3A) encoded from four different genes in the human genome [63,64]. TorsinA has a high degree of sequence homology with torsinB but exhibits different tissue distribution; torsinA is abundant in neuronal tissues, while torsinB is more abundant in non-neuronal tissues [64]. Positional cloning identified that a mutation in *TOR1A*, encoding torsinA with an in-frame deletion of one glutamate residue, causes DYT1 dystonia, which is characterized by involuntary muscular contractions causing twisting movements [63] (Figure 2B). While wild-type torsinA resides in the ER lumen and perinuclear space, the dystonia-causing torsinA mutant is preferentially concentrated in the nuclear envelope [65]. This altered cellular localization of mutant torsinA led Goodchild and Dauer to search for possible nuclear envelope-localized interacting proteins using a cell-based screen. From this candidate screening, the authors found that LAP1 interacts with torsinA. They further identified a transmembrane protein, luminal domain-like LAP1 (LULL1), in the main ER based on sequence homology [62]. 

Unlike other proteins in the AAA+ ATPase family, torsinA is enzymatically inactive unless it interacts with either of two cofactors—LAP1 or LULL1 [66]. Subsequent in vitro reconstitution and biochemical analysis have revealed that LAP1 and LULL1 activate the ATPase of torsinA, accelerating the rate of ATP hydrolysis up to two orders of magnitude. The dystonia-causing mutations interfere with the interaction between torsinA and its cofactors, suggesting that a compromised torsinA ATPase activity causes the disease [67]. Sosa et al. have used X-ray crystallography, electron microscopy, and computer modeling to define the interactions between torsinA and its two cofactors. They showed a heterohexameric ring structure for the torsinA/LAP1 complex and predicted that active site complementation by LAP1 provides a critical residue crucial for torsinA function [68,69]. These rigorous and elegant in vitro experiments consistently point towards a cofactor-induced torsinA activation to be essential for its proper function in different tissues.

Growing evidence shows that defects in torsinA and LAP1 result in various human diseases [64]. While mutations in torsinA cause neurological disorders, mutations in LAP1 have been linked to disorders in multiple tissues. In addition to dystonia-like symptoms, recessive mutations in *TOR1AIP1* that disrupt the LAP1B isoform have been linked to familial cardiomyopathy and muscular dystrophy [70,71,72]. Combined loss of LAP1B and LAP1C causes multisystem disease and death during childhood [73]. Compound heterozygosity in the *TOR1AIP1* that leads to diminished expression of both isoforms causes multisystem alterations in the affected individuals [74] (Table 1). There have been no reports of mutations in *TOR1AIP1* causing NAFLD in humans. Possibly the severe symptoms in other tissues, often leading to an early death, prevent liver abnormalities from having time to develop. The multisystemic alteration may be due to the diverse interactions of LAP1 with proteins in the ER, nuclear membrane, and nucleus, contributing to its roles in neurons and striated muscles. Hence, LAP1 could be a modifier protein in dystonia as well as in muscular dystrophy/cardiomyopathy. These results implicate that LAP1 may have divergent roles besides being a cofactor of torsinA.

### 3.2. The TorsinA/LAP1 Complex in LD Biogenesis and Hepatic Lipid Secretion 

Recently emerging evidence points toward the crucial roles of the torsinA/LAP1 complex at the NE in the regulation of lipid metabolism. The connection of LAP1 and torsinA to lipid metabolism was somewhat unexpected. Shin and colleagues demonstrated the crucial role of LAP1 in striated muscle after establishing a tissue-selective conditional deletion mouse line [55,58,75]. In the course of this study, a liver-specific LAP1 knockout mice (referred to as L-CKO mice) was generated as a control line. While the initial gross examination did not find any abnormalities in the survival or bodyweight of these mice, the authors noticed enlarged hepatocyte nuclei in liver sections. Given the nuclear envelope blebbing phenotypes previously reported in non-neuronal cells with LAP1 deleted [76], the authors performed electron microscopy to find nuclear envelope alterations in L-CKO hepatocytes. They failed to find blebs, but instead, discovered obvious nuclear lipid droplets within the nuclei. Subsequent confocal fluorescent micrographs of isolated hepatocytes from L-CKO mice stained with BODIPY confirmed the presence of LDs both in nuclei and the cytoplasm. Biochemical measurements revealed that livers from L-CKO mice at 6 months of age had significantly increased liver triglyceride (TG) and cholesterol content, indicating hepatic steatosis [24]. 

The abnormal lipid deposition in L-CKO hepatocytes led the authors to test if depletion of emerin or torsinA, two known interacting proteins of LAP1, affects the lipid metabolism similarly. While emerin knockout mice did not show abnormal lipid phenotypes, the liver-specific knockout of *Tor1a* (referred to as A-CKO mice) manifests striking fatty liver phenotypes. Both male and female adult A-CKO mice fed a chow diet exhibited grossly enlarged and whitish livers. Subsequent analysis revealed that hepatic TG and cholesterol was increased nearly 10-fold and 3-fold, respectively, in the livers of 4-month-old A-CKO mice. These mice also demonstrated concomitant drastic reduction in plasma TG and cholesterol content [24]. 

To gain a mechanistic understanding of the marked steatosis phenotypes in these mice, the authors interrogated major hepatic lipid pathways; de novo hepatic lipogenesis, fatty acid oxidation, and lipoprotein secretion. Based on the reduced plasma TG content in both mouse lines, the hepatic VLDL-mediated lipoprotein secretion pathway was suspected. The secretion of TG from hepatocytes is mediated by the formation of VLDL particles, which requires the lipid-carrying protein apoB [77]. The authors performed in vivo VLDL secretion assays to quantify the amount of newly secreted TG and apoB in a certain period of time. These assays revealed a significant decrease in hepatic TG secretion, around 20% in L-CKO mice and nearly 70% in A-CKO mice. Human livers synthesize and secrete only the apoB100 isoform; however, in rodent liver, both apoB100 and apoB48 are produced from a single gene [78]. In agreement with TG secretion defects, the apoB100 secretion was concomitantly reduced in L-CKO and A-CKO mice (approximately 30% and 65%, respectively). In contrast to apoB100, apoB48 secretion was only mildly affected by the depletion of LAP1 or torsinA, suggesting that apoB100-mediated TG secretion is specifically affected. Furthermore, the authors demonstrated that the general hepatic secretion pathway was not affected as the secretion of albumin or apoA1 was unchanged by the depletion of these proteins. Subsequent in vivo de novo lipogenesis assays in both mouse lines did not identify significant alteration; however, in vivo fatty acid oxidation assays identified reduced oxidation in liver samples from A-CKO mice but not in L-CKO mice. These data collectively established a direct role for the torsinA/LAP1 complex in the regulation of apoB100 mediated TG secretion (VLDL secretion). They indicate that a nuclear envelope-localized molecular pathway is key for hepatocyte-autonomous lipid metabolism, defects in which are common features of NAFLD/NASH development.

### 3.3. NASH Development in Chow-Fed Mice with Depletion of LAP1 or TorsinA

Chow-fed mice with hepatocyte-specific deletion of LAP1 or torsinA developed features of NASH with increased fibrosis [24]. Serum alanine aminotransferase (ALT) activity was nearly 4-fold increased in L-CKO mice compared to age-matched control mice (Figure 3A). Histopathological examination of livers from these mice at 18 months of age showed steatosis, lobular inflammation, hepatocyte ballooning (Figure 3B), and increased fibrosis (Figure 3C). The average NAFLD activity score was 3.86, which was significantly higher than that of control mice (1.75). The A-CKO mice demonstrated the same, but accelerated, abnormal liver phenotypes. Livers from 6-month-old A-CKO mice were enlarged and grossly white in color, consistent with severe steatosis. The serum ALT activity was nearly 12-fold higher than that of controls (Figure 3D). Histopathological analysis showed steatosis, lobular inflammation, and hepatocyte ballooning (Figure 3E) and markedly increased pericellular fibrosis (Figure 3F). The average NAFLD activity score for A-CKO mice was 6 as compared to the value for control mice, which was 1.67. The later onset and lower NAFLD activity scores in L-CKO mice, compared to A-CKO mice, are likely due to the presence of LULL1 in the ER, which can partially activate torsinA. In contrast, there is essentially no torsinA activity in the livers of A-CKO mice. 

It is noted that VLDL secretion defects and hepatic steatosis occurred in mice fed a chow diet and in the absence of abnormal glucose homeostasis or obesity. The lack of whole-body metabolic disturbance in L-CKO and A-CKO mice is somewhat contradictory to the frequent association of NAFLD/NASH with metabolic syndrome. However, human genetics studies have identified several genetic insults that predispose individuals to NAFLD/NASH in a liver-autonomous manner, unaccompanied by the features of metabolic syndrome. Human subjects with NAFLD/NASH-susceptible genetic variants in *TM6SF2* do not have insulin resistance or obesity but have steatosis with defects in hepatic VLDL secretion [79,80], similar to the phenotypes observed in L-CKO and A-CKO mice. Individuals with mutations in genes critical to VLDL assembly/secretion, including *MTTP* and *APOB*, are predisposed to steatosis but maintain normal insulin sensitivity [81,82]. Therefore, these results indicate that hepatic steatosis is not a direct driver of insulin resistance in humans. Perhaps, dyslipidemia caused by defective lipid secretion pathways is not linked directly to insulin signaling or other known pathways that can lead to insulin resistance [83,84]. 

## 4. Possible Cellular Mechanisms Connecting Defective Nuclear Envelope Proteins to NAFLD/NASH Pathogenesis

In this section, we will discuss potential cellular mechanisms that can connect the nuclear envelope to the development of NAFLD/NASH based on the abovementioned recent findings. The main focus is on the plausible mechanisms by which torsinA and its activator LAP1 regulate hepatic lipid metabolism and NAFLD/NASH development (summarized in Figure 4). Alteration of these nuclear envelope-localized processes causes robust steatosis phenotypes by disrupting lipid homeostasis directly via hepatocyte-autonomous mechanisms.

### 4.1. VLDL Assembly and Secretion

Human genetic studies on the susceptibility to NAFLD have identified multiple genes involved in VLDL secretion [81,82]. In particular, the *TM6SF2* polymorphism associated with NASH reduces hepatocyte VLDL secretion, confirming that decreased liver TG secretion induces susceptibility to steatosis [80]. The finding that depletion of torsinA or LAP1 at the nuclear envelope causes significant reduction in TG and apoB secretion indicates that decreased torsinA activity is responsible for reduced VLDL secretion and marked hepatic steatosis. The NASH phenotypes observed in these mice at an older age may result from VLDL secretion defects. In this regard, the defective torsinA/LAP1 complex could contribute to the pathogenesis of NASH via dysregulation of the apoB-mediated VLDL secretion.

Hepatic VLDL assembly and secretion is a complex cellular process with the involvement of multiple components, and which of these steps that are regulated by the torsinA/LAP1 complex requires further study. It is possible that the torsinA/LAP1 complex influences the co-translational translocation of apoB100 across the ER membrane. The torsinA activity may be essential for this process and loss of the activity may be associated with a defective translocon function that can lead to nascent apoB100 ubiquitination for proteosomal degradation via the ER-associated degradation (ERAD) pathway [85]. It is also possible that activated torsinA interacts with ER chaperone proteins such as microsomal triglyceride transfer protein (MTP) or protein disulfide isomerase 1, which are required for apoB lipidation and assembly [86,87]. While the torsinA/LAP1 complex is required for VLDL secretion, it should be noted that the hepatic steatosis and NASH resulting from torsinA/LAP1 deletion from hepatocytes is much more severe than that observed in other mice with decreased VLDL secretion. The previously reported *APOB* or *MTTP* knockdown mouse models have around 2–3-fold liver TG increase [88,89]; however, torsinA knockout mice show a nearly 10-fold increase in hepatic TG content. The dramatic lipid retention and NASH phenotypes suggest that there are additional or more fundamental cellular defects in hepatocytes lacking torsinA.

### 4.2. LD Homeostasis and Fatty Acid (FA) Mobilization/Oxidation

LDs are assembled and hydrolyzed in a regulated fashion to supply necessary free fatty acids in hepatocytes [90,91]. Dysregulation of this process is linked to NASH. Patatin-like phospholipase domain containing 3 (PNPLA3) is an LD-associated protein, which participates in the LD lipolytic process. A single-nucleotide variant of *PNPLA3* is strongly associated with NASH progression, emphasizing the importance of the regulation of this lipolysis [84,92]. Defective FA mobilization or oxidation is often linked to NASH [93,94]. Abnormal LD deposition is due to the lack of lipid mobilization to the cellular sites where the lipids will be used, such as to secretory transport vesicles or to the mitochondria for cellular energy generation by β-oxidation. Depletion of torsinA or LAP1 leads to abnormal deposition of LDs in either the ER lumen or cytosol (torsinA deletion) or nucleus (LAP1 deletion), blocking transport of LDs to the correct sites such as the mitochondria (β-oxidation) or the secretion pathway (VLDL secretion). Depletion of torsinA in hepatocytes leads to reduced FA oxidation in mitochondria. It is possible that the loss of the torsinA/LAP1 complex in hepatocytes affects FA-binding protein-mediated FA mobilization to mitochondria [95,96] or the transfer of VLDL from ER to Golgi through COPII-mediated vesicle trafficking [97]. Further studies with proper marker proteins involved in LD biogenesis, FA mobilization, and vesicle transport will be required to conclusively identify the abnormal LD structures observed in hepatocytes lacking torsinA or LAP1. 

### 4.3. Phospholipid Metabolism: Balance between Lipid Storage and Membrane Lipid Synthesis

Phospholipid metabolism influences diverse cellular functions, including membrane remodeling, lipoprotein secretion, lipid droplet formation, lipogenesis, and energy production in mitochondria [98]. In particular, it has been shown that hepatic phosphatidylcholine biosynthesis is essential for hepatic VLDL metabolism in both dietary and genetic mouse models [99,100,101,102]. Phospholipid composition is often altered in NAFLD mouse models and linked to NASH development in humans. Puri and colleagues performed lipidomics and showed a significant decrease in hepatic phosphatidylcholine contents and hepatic lipid subclasses containing arachidonic acid in NAFLD patients [103]. Furthermore, the progress of NAFLD to NASH is characterized by an altered monounsaturated FA and polyunsaturated FA metabolism, perhaps by an impaired oxidation process [104]. Indeed, a recent report demonstrates that oxidized phospholipids accumulate in human and mouse NASH. Moreover, administration of an antibody that neutralizes the oxidized lipid species improves features of NASH in rodent models [9]. 

A recent genetic study of *Drosophila* identified a novel role for torsinA in phospholipid metabolism, which is critical for the balance of lipid storage and membrane lipid synthesis for cell growth. Unlike mammalian cells, which have four different torsins, *Drosophila* has only one, dTorsin, which made direct genetic manipulation possible. The authors found that the expression of dTorsin in the larval fat body was required for the viability of the flies. They also found that the loss of *dTorsin* leads to abnormal hyperactivity of lipin and reduced expression of nuclear PCYT1A, leading to abnormal TG and DAG accumulation and underrepresented PA and phosphatidylcholine species in fat body cells [105]. These findings made the first connection between torsinA and phospholipid homeostasis, which needs to be further validated in the mammalian system.

Subsequent work from the same group reported that lipins are activated in torsinA ΔE mouse brains [106]. In agreement with the results from *Drosophila*, lipidomic analysis of total livers from A-CKO mice showed a marked increase in neutral lipids but a significant reduction in most phospholipid species. The authors did not observe a similar reduction in the lipidomics of total liver samples from L-CKO mice. This may be required to perform a targeted lipidomic analysis with purified nuclear fraction samples from L-CKO hepatocytes as abnormal LD localization was observed within the nucleus. These results from *Drosophila* and mouse indicate that torsinA activity is required for lipid homeostasis, possibly by maintaining the fine balance between lipid storage and membrane lipid synthesis. Whether the function of lipin is altered in hepatocytes lacking torsinA needs to be tested. As lipin-mediated PA phosphatase activity induces TG synthesis in hepatocytes and suppresses membrane phospholipid synthesis, it will be interesting to test if suppressing lipin activity can reduce the hepatic steatosis and resolve the NASH phenotypes in A-CKO and L-CKO mice. 

### 4.4. Lipid Metabolites and Lipid-Mediated Cell Signaling 

Lipid metabolites act as signaling molecules and accumulation of some of lipid metabolites is linked to lipid toxicity-mediated hepatocyte injury and death, which are the hallmark features of NASH. While the accumulation of TG itself is inert or even potentially protective, free FA and non-esterified cholesterol are deleterious to hepatocytes [107,108,109,110]. In particular, some FA-derived lipid moieties, such as DAG, ceramide, and lysophosphatidylcholine, are known as potential lipotoxic lipid species [111,112,113]. For example, DAG plays a role as a second messenger and increased hepatic DAG content activates the protein kinase C (PKC) isoform in the liver, which is linked to hepatic steatosis and hepatic insulin resistance [114,115]. 

The nuclear envelope is known as a cellular signaling node [11]. Numerous studies have reported that nuclear lamins and envelope proteins regulate cell signaling pathways in non-hepatic tissues [116,117,118,119]. Despite the clear connection between defective or absent nuclear envelope proteins and the dysregulation of cellular signaling pathways, it has not been completely delineated how alterations in nuclear envelope proteins initiate or induce these signaling pathways. There is limited information available on how altered nuclear membrane proteins influence cellular signaling pathways in hepatocytes. Selective deletion of A-type lamins in hepatocytes cause aberrant upregulation of the STAT1 signaling pathway [47]. Hepatocyte-specific deletion of torsinA or LAP1 leads to a modest increase in ER stress response and alteration of insulin receptor AKT2-mediated pathways [24]. Targeted lipidomics following subcellular fractionation of isolated hepatocytes from these mice can measure the content of different lipid species and metabolites. Further investigation will be required to examine the direct role of the altered composition of lipid metabolites or lipid-mediated signaling at the hepatocyte nuclear membrane in the regulation of signaling pathways.

### 4.5. Nuclear Receptors and Transcription Factors

One of the promising strategies to treat NASH is targeting nuclear receptors or transcription factors crucial for lipid metabolism and fibrosis [10,120]. Two NASH drugs that have entered phase III development target nuclear receptors. The farnesoid X receptor (FXR) is a bile acid receptor and its agonist, obeticholic acid, improves insulin sensitivity and exhibits anti-inflammatory and antifibrotic effects in mouse models of NASH and human tissues [121,122]. The peroxisome proliferator-activated receptor (PPAR)α/ δagonist elafibranor shows resolution of NASH without worsening fibrosis [123]. Chemokine receptor 2 (CCR2) and CCR5 antagonist, cenicriviroc, are currently being used in a phase II study for NASH with fibrosis [124,125]. 

The nuclear lamins and lamin-associated proteins bind to and potentially modulate the localization or activities of transcription factors involved in regulating lipid metabolism [126]. It has been reported that FPLD-causing mutations in the gene encoding lamin A alter the localization and activity of SREBP1 [127]. In addition, overexpression of lamin B1 also reduces the mRNA level of *SREBP1* and *2* [52]. Overexpression of the precursor form of lamin A impairs preadipocyte differentiation and reduces expression of the adipogenic transcription factor PPARγ [128]. It is possible that the torsinA/LAP1 complex in the inner nuclear membrane may interact with nuclear proteins implicated in lipid homeostasis. Gene expression profiling combined with proteomic analysis will identify these nuclear factors. 

### 4.6. Epigenetic Regulation

An accumulating body of evidence indicates that epigenetic alteration is linked to NAFLD and the development of HCC. Epigenetic modifications are caused by alteration in DNA methylation, histone modification, chromatin remodeling, and RNA-mediated mechanisms. Data suggest that insults such as nutritional or oxidative stress can lead to stable epigenetic alteration that can influence susceptibility to NAFLD and hepatocyte tumorigenesis [129,130,131,132]. The nuclear lamina influences genome organization and the gene expression network. Lamin A/C interacts with chromatin directly in a cell type-specific manner at the genomic regions called *LMNA*-associated domains (LADs) [133,134,135]. Lamin A/C also regulate chromatin by interacting with key epigenetic regulators including the proteins of the polycomb group, which can modulate the myogenic transcriptional program during muscle differentiation [136]. It is also known that lamin-associated inner nuclear membrane proteins bind to various chromatin modifier proteins [137]. It is possible that LAP1 in the nuclear envelope directly or indirectly interact with chromatin or chromatin-modifying proteins to control gene expression. Altered epigenetic regulation due to the disruption of the torsinA/LAP1 complex in hepatocytes could lead to dyslipidemia and NASH development.

## 5. Conclusions

In conclusion, we have described emerging evidence providing links between nuclear membrane proteins and lipid metabolism in various systems. To our knowledge, no prior genome-wide association studies have robustly linked the genes encoding nuclear lamins and nuclear envelope proteins to NAFLD/NASH cases. More studies with increased statistical power will be required to detect rare polymorphisms existing in these genes. Direct sequencing of genes encoding these proteins should be considered in individuals with NAFLD/NASH. One such study with a limited subject number identified variants in the LAP2 encoding gene among a cohort of twins and siblings with NAFLD [48]. Further investigation will be required to determine whether the defects in the nuclear envelope are the direct pathogenic drivers or secondary responses to the cellular stresses contributing to NASH development. The detailed molecular mechanisms by which nuclear lamins and membrane-associated proteins regulate lipid metabolism remain to be elucidated. This line of research will not only provide fundamental knowledge of the nuclear envelope biology but also present the nuclear envelope as a new subcellular site for active lipid regulation and contribute to the understanding of the pathogenesis of NASH. 

## Figures and Tables

**Figure 1 biology-09-00338-f001:**
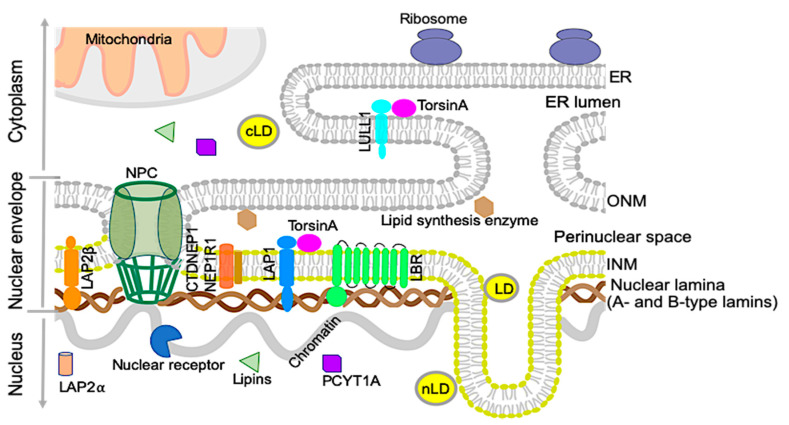
The organization of the nuclear envelope and endoplasmic reticulum (ER). Schematic diagram of the nuclear envelope and ER showing the nuclear membranes, nuclear lamina, and nuclear pore complex (NPC). The nuclear lamina contains A- and B-type lamins. The outer nuclear membrane (ONM) is contiguous with the ER, containing lipid synthesis enzymes and ribosomes. The ER lumen is directly continuous with the perinuclear space. Selected proteins, such as lipid synthesis enzymes concentrated in the ER and inner nuclear membrane (INM), are shown along with torsinA in the perinuclear space. Lamina-associated polypeptide 1 (LAP1) is shown interacting with lamins and torsinA. Two isoforms of LAP2 are shown in the nucleoplasm (LAP2α) and INM (LAP2β). Some of the nuclear receptors that regulate gene expression are known to interact with nuclear lamins or lamin-associated proteins. Some enzymes involved in phospholipid biogenesis (CTDNEP1/NEP1R1, lipins, and PCYT1A) are localized in the nucleus or NE. TorsinA interacts with LAP1 at the nuclear envelope (NE) and luminal domain-like LAP1 (LULL1) in the main ER. Lipid droplets are mainly synthesized in the ER and present in the cytosol as cytosolic lipid droplets (cLDs). Certain types of cells contain nuclear lipid droplets (nLDs) within the nucleus. Mitochondria and ribosomes are localized in the cytoplasm.

**Figure 2 biology-09-00338-f002:**
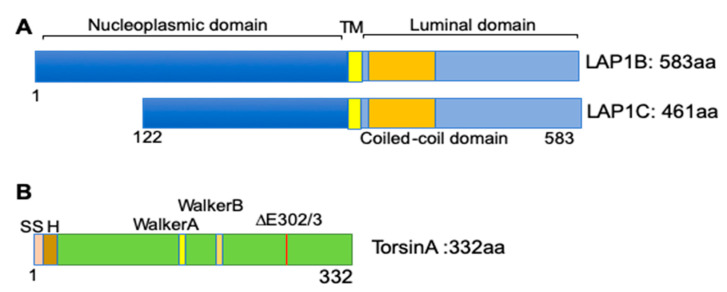
Protein structure of human LAP1 and torsinA. Schematic diagrams showing the known protein domains of human LAP1 isoforms and torsinA. (**A**) Diagrams of human LAP1B and LAP1C; nucleoplasmic (blue), transmembrane (TM) (yellow), coiled-coil (orange), and luminal domains (light blue) are shown. (GenBank database *TOR1AIP1* (*LAP1B*: NM_015602.4, *LAP1C*: AAH23247.1); UniProt ID: Q5JTV8) [56]. (**B**) Diagram of the human torsinA protein structure (GenBank database *TOR1A*: NM_000113.3; UniProt ID: O13656) [63], showing signal sequence (SS), hydrophobic domain (H), Walker A: ATP-binding motif, Walker B: ATP hydrolysis motif, and ΔE302/3, site of mutation causing DYT1 dystonia.

**Figure 3 biology-09-00338-f003:**
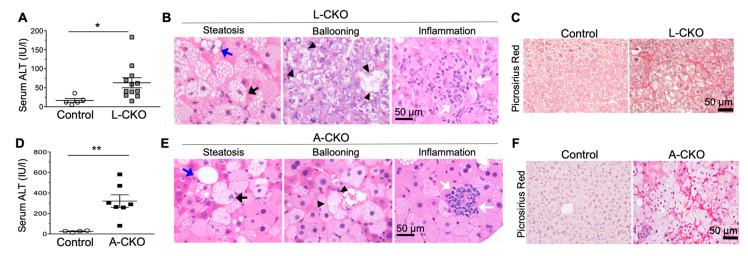
Nonalcoholic steatohepatitis (NASH) features and fibrosis in chow-fed L-CKO mice and A-CKO mice. (**A**) Serum alanine aminotransferase (ALT) activity in L-CKO mice at 12–18 months of age (*n* = 5–12 mice per group). * *p* < 0.05, by Student’s *t* test. (**B**) Representative photomicrographs of H&E-stained liver sections from L-CKO mice at 18 months of age; scale bar: 50 µm. Examples of steatosis, hepatocyte ballooning, and inflammation are shown. Macrovesicular steatosis is indicated by blue arrows and microvesicular steatosis is indicated by black arrows. Arrowheads indicate hepatocytes with ballooning degeneration. White arrows indicate inflammatory cells. (**C**) Picrosirius Red-stained liver sections showing increased fibrosis in L-CKO mice at 18 months of age. (**D**) Serum ALT activity in A-CKO mice at 6 months of age (*n* = 4–7 mice per group). ** *p* < 0.01, by Student’s *t* test. (**E**) Representative H&E-stained sections of livers from A-CKO mice at 6 months of age. Examples of steatosis, hepatocyte ballooning, and inflammation are shown; Macrovesicular steatosis is indicated by blue arrows and microvesicular steatosis indicated by black arrows. Arrow heads indicate hepatocytes with ballooning degeneration. White arrows indicate inflammatory cells. (**F**) Picrosirius Red-stained liver sections showing increased fibrosis in A-CKO mice at 6 months of age. (Reproduced with permission from Shin et al., 2019 [24]).

**Figure 4 biology-09-00338-f004:**
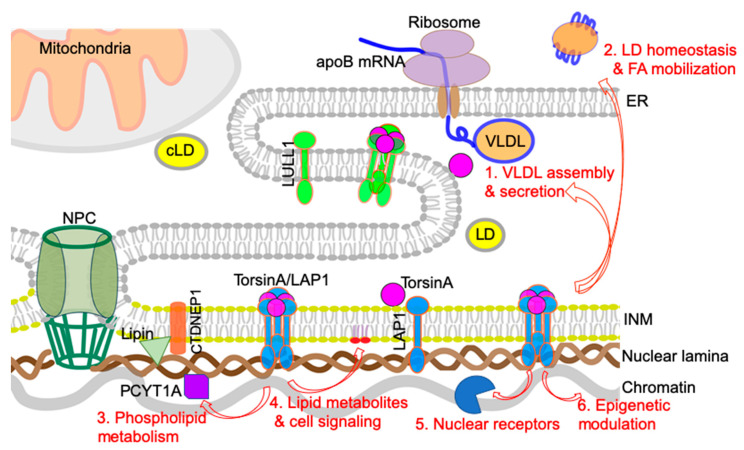
Possible cellular mechanisms that connect defective torsinA/LAP1 complexes at the nuclear envelope to NAFLD/NASH pathogenesis. TorsinA interacts and from a heterohexameric structure with LAP1 in the nuclear envelope or with LULL1 in the ER. Red arrows indicate the plausible cellular pathways that torsinA/LAP1 are involved in in hepatocytes. See text for detailed descriptions.

**Table 1 biology-09-00338-t001:** Mutations in *TOR1AIP1* associated with human disease.

References	Reported Year	No. of Affected Individuals	Mutation in *TOR1AIP1*	Resultant LAP1 Protein	Phenotypes
Kayman-Kurekci et al. [70]	2014	3	c.186delG/c.186delG	p.E62fs*25 (truncation at 83 aa of LAP1B but intact LAP1C)	muscular dystrophy, joint contracture, cardiomyopathy
Dorboz et al. [71]	2014	1	c.1448A > T/c.1448A > T	p.E482A (E to A change in both LAP1 isoforms)	cerebellar atrophy, dystonia, cardiomyopathy, early death
Ghaoui et al. [72]	2016	2	c.127delC/c.1181T > C	p.P43fs*15/p.L394P (truncation at 58 aa of LAP1B, L to P change in both LAP1 isoforms)	muscular dystrophy, cardiomyopathy
Fichtman et al. [73]	2019	7	c.961C > T/c. 961C > T	p.R321* (truncation at 321 aa of both LAP1 isoforms)	multisytemic abnormalities, early death
Lessel et al. [74]	2020	2	c.945_948delCAGT/c.1331G > C	p.Q315fs*9/p.R444P (truncation at 315 aa and R to P changes in both LAP1 isoforms)	congenital hearing loss, developmental delay, brain abnormality

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
