# Peer review of "The Nuclear Envelope in Lipid Metabolism and Pathogenesis of NAFLD"

_biology, 2020, doi:10.3390/biology9100338_

Round 1

Reviewer 1 Report

In this review, the authors summarise the role that the nuclear envelope, and its associated proteins, play in lipid metabolism and the pathogenesis of NAFLD/NASH. The review, in particular, covers findings from the authors recent paper Nuclear envelope-localized torsinA-LAP1 complex regulates hepatic VLDL secretion and steatosis. This is a novel topic, which the authors highlight, has currently been difficult to investigate. The review is well written, covering a good introduction, some of the key proteins that may be implicated, and the future questions that need to be addressed. The review is of great interest and worthy of publication, pending some minor requests.

  • The major point that needs addressing is in regards to tissue specificity. Firstly, are the proteins of interest ubiquitously expressed, and are the different isoforms tissue specific (how many isoforms are there). What is the tissue expression profile for these proteins/genes. Are the phenotypes mentioned (in particular in table 2), due to mutations in these proteins in specific tissues or are they global. In the incidences of these phenotypes in table 2, is there any correlation or association with fatty liver.

  • The results suggest alterations in de novo lipogenesis without detriments to glucose tolerance or adiposity. Is it known whether these mutations alone in humans can lead to NAFLD, or are other insults, such as excess calories required.

  • Do the authors have any information on nLD versus cLD in regards to the severity or likelihood of NAFLD progressing to NASH or HCC.  

  • There is no mention in the introduction of obesity and its contribution/role to the prevalence of NAFLD/NASH.

  • Further GWAS studies, as mentioned in the conclusion are of interest.

Author Response

Author’s reply to the Reviewer 1

We appreciate the reviewer’s helpful and constructive comments. We have responded to the comments point by point and incorporated our changes accordingly in our revised manuscript.

Reviewer 1

In this review, the authors summarise the role that the nuclear envelope, and its associated proteins, play in lipid metabolism and the pathogenesis of NAFLD/NASH. The review, in particular, covers findings from the authors recent paper Nuclear envelope-localized torsinA-LAP1 complex regulates hepatic VLDL secretion and steatosis. This is a novel topic, which the authors highlight, has currently been difficult to investigate. The review is well written, covering a good introduction, some of the key proteins that may be implicated, and the future questions that need to be addressed. The review is of great interest and worthy of publication, pending some minor requests.

  • The major point that needs addressing is in regards to tissue specificity. Firstly, are the proteins of interest ubiquitously expressed, and are the different isoforms tissue specific (how many isoforms are there). What is the tissue expression profile for these proteins/genes.

à As for LAP1, the ubiquitous expression and its isoforms were mentioned in section 3.1: “ The rodent LAP1 encoding gene, Tor1aip1, is expressed in three isoforms (LAP1A, LAP1B and LAP1C) with apparent molecular masses of 75 kDa, 68 kDa and 55 kDa [53]. These isoforms are expressed in most tissues in a differential pattern dependent on the developmental stages [55]. In humans, at least two isoforms are expressed from the TOR1AIP1 gene [56].”

à We also added more explanations about different torsin members and tissue expression patterns. “There are four torsins (torsinA, torsinB, torsin2A and torsin3A) encoded from four different genes in the human genome [63,64]. TorsinA has a high degree of sequence homology with torsinB, but exhibits different tissue distribution; torsinA is abundant in neuronal tissues while torsinB is more abundant in non-neuronal tissues [64]”

Are the phenotypes mentioned (in particular in table 2), due to mutations in these proteins in specific tissues or are they global. In the incidences of these phenotypes in table 2, is there any correlation or association with fatty liver.

à It has to be read as Table 1 (there is no table 2):  “While mutations in torsinA cause neurological disorders, mutations in LAP1 have been linked to disorders in multiple tissues”, We have added additional comments regarding fatty-liver related phenotypes in the last paragraph of section 3.1:

“There has been no reports of mutations in TOR1AIP1 causing NAFLD in humans. Possibly the severe symptoms in other tissues, often leading to an early death, prevent liver abnormalities from having time to develop”.

  • The results suggest alterations in de novo lipogenesis without detriments to glucose tolerance or adiposity. Is it known whether these mutations alone in humans can lead to NAFLD, or are other insults, such as excess calories required.

àAs we stated in the manuscript (see section 3.2), “ Subsequent in vivo de novo lipogenesis assays in both mouse lines did not identify significant alteration” , the hepatocyte-specific deletion of LAP1 or torsinA mice exhibit NORMAL rates of hepatic de novolipogenesis, as well as normal glucose tolerance or fat body mass.

  • Do the authors have any information on nLD versus cLD in regards to the severity or likelihood of NAFLD progressing to NASH or HCC.  

à This is an excellent question. At this moment, it is completely unknown. We added a sentence in section 2.1

“The differential impacts of nLDs and cLDs on the NAFLD pathogenesis and progression are unknown”

  • There is no mention in the introduction of obesity and its contribution/role to the prevalence of NAFLD/NASH.

àNow included in the introduction.

  • Further GWAS studies, as mentioned in the conclusion are of interest.

à We also agree that further GWAS studies are of interest.

Reviewer 2 Report

In the submitted review manuscript entitled, “The nuclear envelope in lipid metabolism and pathogenesis of NAFLD,” the authors nicely review the literature on this topic, including their recent publication describing the role of LAP1 in NAFLD/NASH. In the context of that recent publication and prior articles implicating lamin proteins and LAP2 in NASH, this review is quite timely. Additionally, it is well-written and comprehensive, and it points toward important avenues of future study. Overall this is an important snapshot of the state of the field.

Major Comments: none

Minor Comments:

  1. The section of the review summarizing the authors’ recent JCI article could be condensed.
  2. There are a couple of run-on sentences that should be edited.
  3. In terms of ways to improve the manuscript, I would suggest trying to divide attention to nuclear envelope proteins and their contributions to NAFLD/NASH in a more even fashion rather than devoting so much of the manuscript to torsinA and LAP1. I understand that this is the authors' primary pathway of interest and they wish to highlight their recent excellent publication on this topic, but it is probably not necessary to go through the results of that study in such detail here.
  4. And in particular, I would not necessarily have included primary data from that study (better to send readers to the full primary article). However, in terms of the overall quality of the writing and accuracy of the content, it is well done. I hope these comments are helpful.

Author Response

Author’s reply to the Reviewer 2

We appreciate the reviewer’s helpful and constructive comments. We have responded to the comments point by point and incorporated our changes accordingly in our revised manuscript.

Reviewer 2

In the submitted review manuscript entitled, “The nuclear envelope in lipid metabolism and pathogenesis of NAFLD,” the authors nicely review the literature on this topic, including their recent publication describing the role of LAP1 in NAFLD/NASH. In the context of that recent publication and prior articles implicating lamin proteins and LAP2 in NASH, this review is quite timely. Additionally, it is well-written and comprehensive, and it points toward important avenues of future study. Overall this is an important snapshot of the state of the field.

Major Comments: none

Minor Comments:

  1. The section of the review summarizing the authors’ recent JCI article could be condensed.

à We have now removed some of the sentences describing results from the recent JCI article. In particular, we removed the methodology description on in vivo secretion assay, which is well described in the original article.

  1. There are a couple of run-on sentences that should be edited.

à We have now fixed run-on sentences that we could detect in the text.

  1. In terms of ways to improve the manuscript, I would suggest trying to divide attention to nuclear envelope proteins and their contributions to NAFLD/NASH in a more even fashion rather than devoting so much of the manuscript to torsinA and LAP1. I understand that this is the authors' primary pathway of interest and they wish to highlight their recent excellent publication on this topic, but it is probably not necessary to go through the results of that study in such detail here.
  2. And in particular, I would not necessarily have included primary data from that study (better to send readers to the full primary article).

à We appreciate the reviewer’s comments, however, we decided to leave Figure 3 to highlight the NASH features of our mouse models, which is the only figure that we took from the original paper.

However, in terms of the overall quality of the writing and accuracy of the content, it is well done. I hope these comments are helpful.

à We thank the reviewer for his/her appreciation of our manuscript. The comments were constructive to improve our manuscript.